# Bilateral Total Knee Arthroplasty (TKA) in a One-Stage Procedure Versus Two-Stage Procedure: A Retrospective Study

**DOI:** 10.3390/healthcare12181902

**Published:** 2024-09-23

**Authors:** Giada Accatino, Alessandra Monzio Compagnoni, Federico Alberto Grassi, Alberto Castelli, Gianluigi Pasta, Francesco Benazzo, Stefano Marco Paolo Rossi, Eugenio Jannelli

**Affiliations:** 1Department of Clinical, Surgical, Diagnostic and Pediatric Sciences, University Study of Pavia, 27100 Pavia, Italy; 2Orthopedics and Traumatology Clinic, IRCCS Policlinico San Matteo Foundation, 27100 Pavia, Italy; 3Robotic Prosthetic Surgery Section–Sports Traumatology Unit, U.O.C Orthopedics and Traumatology Poliambulance Foundation, 25121 Brescia, Italy

**Keywords:** bilateral, total knee arthroplasty, TKA, one stage, two stage, osteoarthritis

## Abstract

**Background/Objectives:** This retrospective study aims to evaluate the safety and functional outcomes of simultaneous bilateral total knee arthroplasty (TKA) compared to two-stage bilateral TKA procedures. Osteoarthritis is the leading cause of knee joint replacement globally, so we sought to determine whether the one-stage approach could be deemed non-inferior to the two-stage method in terms of perioperative complications and postoperative recovery. **Methods:** We analyzed two cohorts: 43 patients underwent one-stage bilateral TKA, while 66 patients received two-stage TKA. The data collected included demographic details, comorbidities, postoperative complications, and functional outcomes assessed by the Oxford Knee Score (OKS), European Quality of Life, and Visual Analogue Scale scores at preoperative and five years postoperative follow-ups. **Results:** The results showed statistically significant improvements in quality-of-life measures for both groups, with the one-stage group exhibiting greater enhancement in OKS (*p* < 0.05). Complication rates were similar across both procedures, with no significant differences in infection rates or other adverse events. **Conclusions:** This study suggests that the one-stage approach to treat bilateral knee arthritis could reduce subjective stress and healthcare costs, presenting a safe alternative for appropriately selected patients. However, the findings highlight the necessity of considering patients’ overall health status when planning surgical interventions. Further research involving larger populations is recommended to validate our conclusions.

## 1. Introduction

The knee is the joint most affected by osteoarthritis, accounting for almost four-fifths of patients worldwide [1]. Osteoarthritis is a chronic irreversible condition that progressively becomes resistant to conservative treatments, such as pharmacotherapy and physiotherapy. Joint replacement surgery remains the only definitive approach to control symptoms and improve patient’s quality of life [2]. Today, total knee arthroplasty (TKA) is a routine procedure in orthopedic surgery to restore function in cases of severe osteoarthritis or systemic diseases. Diagnosis is mainly radiographic. The stage of knee osteoarthritis can be determined through one of the most widely used radiographic classifications: the Kellgren–Lawrence classification, which describes osteoarthritic joints in four stages of severity based on morphostructural changes [3].

TKA indications have been rising constantly in recent decades. Increased life expectancy, profound differences in terms of functional demand, especially in elderly patients, and advancements in surgical and anaesthesiologic techniques are just some of the main factors that have led to an inexorable growth in these implants. The incidence rate of bilateral osteoarthritis is 5%. One-third of patients undergoing knee prosthesis surgery require a contralateral knee prosthesis surgery in the following years [4].

There are two options for patients that require bilateral knee replacements: a procedure defined as “one-stage”, in which the patient undergoes the replacement of both knees with one hospitalization and surgical procedure, and a procedure defined as “two-stage”, in which the patient is hospitalized twice to undergo two separate surgical procedures over time. The literature is not in agreement on which of the two procedures is safer and more functional for the patient’s recovery: perioperative morbidity and mortality rates in simultaneous bilateral knee replacement surgery remain a debated aspect due to the fact that an increase in perioperative complications, such as deep vein thrombosis, pulmonary embolism, cardiac and neurological complications, and wound complications has been reported, with a related higher need for intensive care unit admissions. At the same time, other studies have reported an increased incidence of complications in multi-stage procedures [5,6,7].

The aim of our study is to compare whether the one-stage procedure can be considered non-inferior to the procedure involving two separate surgeries in terms of functional outcomes. Furthermore, perioperative complications were evaluated.

## 2. Materials and Methods

This study evaluates patients undergoing bilateral total knee replacement surgery from January 2010 to November 2019. This is a retrospective study and compares two populations: a cohort undergoing bilateral one-stage surgery and a control cohort undergoing bilateral two-stage surgery with less than 12 months between the two procedures. The surgeries were performed by the same operator. Data were collected using the computerized system of the hospital where the surgeries were performed.

### 2.1. Population

All patients had an indication for bilateral surgical treatment from the outset, with a Kellgren–Lawrence score of 3 or 4 (Figure 1), without previous knee replacement surgeries. Patients with unicompartmental knee replacements, complicated prostheses, secondary arthritis due to systemic or post-traumatic disorders, and with coagulation disorders were excluded.

Regarding the postoperative period, patients began passive and active joint mobilization exercises of the knee and isometric strengthening exercises of the quadriceps femoris on the first day; on the second day, patients began protected assisted ambulation using anti-thromboembolic stockings. All patients underwent a rehabilitation cycle lasting an average of three weeks at a specialized institution.

### 2.2. Surgical Technique

A single operator performed all surgeries, without a tourniquet, using the same surgical technique with a mini-tri-vector approach. The technique involved a distal femoral cut with intramedullary alignment, a proximal tibial cut perpendicular to the mechanical axis of the tibia, gap balancing with the knee in extension, and chamfer cuts with external rotation based on anatomical axes such as the trans-epicondylar axis and Whiteside lines. In all patients, the patella was resurfaced [8]. Cementation was performed after pulsatile lavage, without inducing limb ischemia, with cement placed on both the tibial bone and plate, including the keel. No sclerotic bone perforation was performed. All implants used were the Persona Zimmer Biomet PS. The average surgical time for the one-stage procedure was 90 min (range 75–100 min). The average surgical time for the two-stage procedure was 170 min (150–190 min). The average surgical time is calculated from the time of the preparation of the surgical field until suturing and medication.

Antibiotic prophylaxis with cefazolin was used. All patients underwent radiographs (Figure 2) in the immediate postoperative period.

A total of 97 one-stage bilateral knee replacement surgeries were selected. Out of this group, 51 patients were excluded from this study for the following reasons: 43 patients already had a unicompartmental knee prosthesis at the time of bilateral TKA indication; 6 patients had autoimmune diseases (4 had rheumatoid arthritis, 2 had psoriatic arthritis); 2 patients had coagulopathy (hemophilia); 3 patients were lost to follow-up. The case group therefore consisted of 43 units.

From the total of 128 patients who underwent bilateral knee replacement surgeries in two stages, a final sample of 66 patients was selected. Exclusion criteria were applied as follows: 60 patients did not receive initial recommendation for bilateral treatment; 4 patients had rheumatoid arthritis; 2 patients had coagulopathy (hemophilia); 4 patients were lost to follow-up; 1 patient died from causes unrelated to the treatment under investigation.

### 2.3. Follow-Up

All patients were followed up regularly at our clinic at 1 month, 3 months, and 6 months from the unique surgery for the case group and from the last surgery for the control group. The final evaluation was conducted at 5 years. The first parameter evaluated was pain before and after surgery; it was assessed through the Visual Analogue Scale, which is a validated and subjective measure for acute and chronic pain recorded by making a handwritten mark on a line that represents a continuum between “no pain” and “worst pain” [7]. We chose to use Patient-reported outcome measures (PROMs) to evaluate the degree of well-being and satisfaction of the patients, subjecting them to questionnaires related to their condition before the surgery and 5 after years. The Oxford Knee Score (OKS) consists of 12 questions about an individual’s level of function, activities of daily living, and how they have been affected by pain over the preceding four weeks [8]. The European Quality of Life EQ-5D is a widely used, standardized, preference-based generic measure of health-related quality of life through five dimensions (i.e., mobility, self-care, usual activities, pain/discomfort, anxiety/depression) [9]. The Forgotten Joint Score (FJS) is composed of 12 items measuring the patient’s ability to forget the presence of an artificial joint in their daily life [10]. The Knee Society Score (KSS) is a validated and responsive method for assessing patients’ satisfaction, functional activities, and expectations and clinical conditions by a physician [11]. During the interim follow-ups, questions about any postoperative complications such as allergies, adverse events, urinary problems, bleeding, or wound dehiscence were asked.

Perioperative variation in hemoglobin, as well as whether transfusion had occurred, was also assessed for each patient by means of the computerized blood test collection system and from anaesthesiology records. The threshold values for transfusions were based on the Patient Blood Management (PBM) guidelines: hemoglobin levels below 7 mg/dL, hemoglobin levels below 8 mg/dL in cardiac or symptomatic patients.

Any adverse events such as the onset of any infections were evaluated and reported.

### 2.4. Statistical Analysis

The data obtained in this study were analyzed using Microsoft 365 Excel software V2021. The values of the two populations were assessed as homogeneous, so a two-tailed *t*-test was used for independent variables in the case–control comparison, whereas a paired-variables *t*-test was used in the pre- and post-operation comparison. A *p* value < 0.05 was considered statistically significant.

## 3. Results

### 3.1. Demographic Analysis

The final population consisted of 43 patients undergoing one-stage bilateral knee replacement surgery (case group) and 66 patients undergoing two-stage bilateral knee replacement surgery (control group).

The average age for the case group was 70.2 years, with a range of 42 to 85 years. The gender distribution showed that 33% were male and 67% were female.

The average age for the control group was 64.2 years, with a range of 48 to 75 years. The gender distribution showed that 45% were male and 55% were female (Table 1).

The postoperative follow-up period ranged from a minimum of 12 months to a maximum of 94 months, with an average value of 59.2 months for the case group, whereas the control group follow-up ranged from a minimum of 12 months to a maximum of 78 months, with an average value of 52.8 months.

The case group presented the following comorbidities: eight patients had diabetes mellitus, twenty-seven had arterial hypertension, seven had coronary artery disease, five had anxiety and depression syndrome, three had liver diseases, and three had kidney diseases. In terms of gonalgia before surgery, 18 patients had had painful symptoms for less than 1 year, 37 for a period between 1 and 5 years, 19 for a period between 6 and 10 years, and 10 for a period exceeding 10 years. The control group presented the following comorbidities: six patients had diabetes mellitus, twenty-seven had arterial hypertension, seven had coronary artery disease, one had anxiety and depression syndrome, and one had liver diseases. In terms of gonalgia before surgery, 9 patients had had painful symptoms for less than 1 year, 37 for a period between 1 and 5 years, 12 for a period between 6 and 10 years, and 10 for a period exceeding 10 years (Table 2).

### 3.2. Results Analysis

#### 3.2.1. Oxford Knee Score

The average Oxford Knee Score in the case group before surgery was 27.4 (range 24–32), whereas the average score at 5 years of follow-up was 43.1 (range 34–47). The average increase from preoperative to postoperative values was 15.7 (*p* value < 0.05).

The average Oxford Knee Score before surgery in the control group was 29.3 (range 25–34) and at the 5-year follow-up, it was 44.2 (range 34–46). The average increase in the score was 14.9 (*p* value < 0.05) (Figure 3).

Both groups showed a statistically significant improvement following bilateral knee replacement surgery. However, the control group had a slightly higher score of 1.1 (*p* value = 0.63).

#### 3.2.2. EQ-5D

The average EQ-5D value in the case group before surgery was 0.35 (range 0.174–0.433), whereas the average value at 5 years of follow-up was 0.9 (range 0.274–1). The average increase from preoperative values to postoperative values was 0.55 (*p* value = 0.042).

The average EQ-5D value preoperatively in the control group was 0.5 (range 0.411–0.620) and at the 5-year follow-up, it was 0.92 (range 0.497–1). The average increase in the score was 0.420 (*p* value = 0.3) (Figure 4).

Only the case group showed a statistically significant improvement in quality of life at 5 years of follow-up, even though the value was just 0.02 less than the control group’s value (*p* value = 0.74).

#### 3.2.3. VAS Score

The average VAS score in the case group before surgery was 91.3 (range 60–100), whereas the average score at 5 years of follow-up was 53.3 (range 40–60). The average decrease from preoperative to postoperative values was 28 (*p* value > 0.05).

The average VAS score preoperatively in the control group was 90.5 (range 60–100) and at the 5-year follow-up, it was 53 (range 40–60). The average decrease in the score was 37.5 (*p* value < 0.05) (Figure 5).

The decrease in VAS score in the case group was 0.8 less than in the control group (*p* value = 0.65).

#### 3.2.4. Forgotten Joint Score

The average value of the Forgotten Joint Score at 5 years of follow-up in the case group was 70.8, whereas in the control group it was 70.4, with an identical range in both groups (25–100). Comparing the two groups, no statistically significant difference was found (*p* value = 0.59) (Figure 6).

#### 3.2.5. Knee Society Score

The average value of the Knee Society Score at 5 years of follow-up in the case group was 80.3 (range 71–100), whereas in the control group, it was 80.95 (range 64–100). The case group therefore showed an average value 0.65 lower than the control group (*p* value = 0.38) (Figure 6).

### 3.3. Complications

During the days following the surgical intervention and intermediate follow-ups, any complications that arose were recorded.

In the case group, the complications were as follows: two patients experienced a urinary infection due to prolonged catheterization; two patients developed a hematoma due to excessive bleeding. Only one adverse drug reaction was reported.

In the control group, the complications were as follows: one patient experienced a urinary infection and three patients developed a hematoma.

### 3.4. Reintervention

No revision surgery was necessary in the case group. In the control group, one patient underwent surgical revision for aseptic loosening 3 years after the initial surgery, which required a one-stage procedure.

### 3.5. Hemoglobin

The average decrease in hemoglobin levels per patient was 3.8 g/dL (range 1.6–6.8 g/dL) in the case group. Transfusion of a bag of leukodepleted packed red blood cells was required for 20.9% (9 out of 43) of hospitalized patients. None of the patients required more than one dose of transfusion.

The average decrease in hemoglobin levels per patient was 3.56 g/dL (range 1.4–6.5 g/dL) in the control group. Transfusion of a bag of leukodepleted packed red blood cells was required for 12.4% (12 out of 66) of hospitalized patients. None of the patients required more than one bag of transfusion.

The difference in the mean of transfusion of the two groups was not statistically significant (*p* value = 0.27).

## 4. Discussion

Today, many surgeons are still hesitant to offer simultaneous bilateral knee replacement surgery to patients with arthritis due to concerns about complications.

The results are conflicting: while the reduction in hospitalization time, subsequent rehabilitation, and cost containment for the National Health Service support the one-stage procedure, some authors discourage this procedure due to an increased risk of complications and mortality [12]. Other studies, however, have not found any difference in the rate of postoperative medical complications (myocardial infarction, pulmonary embolism, deep vein thrombosis, cardiovascular accidents, gastrointestinal bleeding, and pneumonia) between simultaneous and multi-stage bilateral total joint arthroplasty [13].

In lights of these conflicting data, this study aimed to provide a comprehensive overview of the clinical and functional outcomes of the two procedures at a medium-term follow-up.

Based on our results, there was a statistically significant increase in clinical outcomes (OKS) in both groups. As for functional outcomes (EQ-5D and VAS), both groups had an improvement, but only in the case group for EQ-5D and in the control group for VAS were there statistically significant increases.

Quality of life assessed through the EQ-5D score improved in both the one-stage and two-stage groups, reaching a value equal to or above 0.9. However, only patients who underwent simultaneous bilateral surgery showed a statistically significant improvement in quality of life at 5 years of follow-up, although the score value in this group was 0.02 lower (*p* value = 0.74) compared to patients who underwent two separate procedures.

At 5 years of follow up, patients in the one-stage group reported less pain compared to the other group, despite a significant improvement in pain symptoms in both groups. In terms of patient perception of the prosthetic implant in daily life, there were no statistically significant differences between the two groups, which reached an FJS of about 70. According to the KSS, too, the two groups were comparable, with scores of 80, without statistically significant differences in the improvement of general joint functionality.

However, when analyzing the raw data, comparable values were observed in all scores. Therefore, the greater improvement seen in patients treated with the one-stage procedure could be associated with the greater impact that severe bilateral arthritis had on the patients’ lives [14]. This fact is consistent with studies in the literature showing greater increases in terms of range of motion and functional scores in patients undergoing a single surgical procedure.

No difference in terms of medical complications, such as infection, was noted between the groups, although none of the patients treated had a high preoperative risk. Regarding infections, some authors reported a higher infection rate in the one-stage method, whereas others did so in the two-stage one, probably due to double exposure and longer overall hospitalization [15,16].

However, the one-stage intervention group had an average higher blood loss than the two-stage group, but this was not statistically significant. A higher percentage of patients with one-stage surgery were transfused, and these results are in line with the literature. Although there were no actual adverse events in our report, we should remember that increased bleeding peri- and postoperatively, and the need for transfusions, can still represent a negative prognostic factor [13].

In the literature, prior meta-analyses have noted a higher rate of complications associated with simultaneous total knee replacement in comparison to unilateral TKA; for example, a higher risk of pulmonary embolism [17,18,19].

In addition, the series should include separate assessments of bilateral cases, because they have better survival than unilateral knees. A long delay between the two knee replacements could have an impact on the survival of both implants [20].

During follow-up, no cases of postoperative stiffness were reported in our studies. These data could be correlated to the time between the two surgeries: as is the recommendation of the Consensus Conference in Bilateral Total Knee Arthroplasty Group [10], our patients underwent the second surgery after a minimum of 3 months. This observation could be associated with a higher inflammatory response in the initial 3 months after a surgery [21,22,23]. In a meta-analysis reported by Restrepo et al. [19], the risk of pulmonary embolism was higher in simultaneous bilateral TKA than in unilateral TKA. On the other hand, Richardson et al. [24] and Poultsides et al. [25] did not find a significant difference in this complication in groups that underwent a simultaneous and staged procedure, in agreement with our results [22].

In our study, no cases of mortality were reported. In the literature, the impact of the two different types of surgery on mortality rates is controversial. This may in part be attributed to the relatively rare occurrence of mortality, which could be a bias [26,27].

### Study Limitations

This study has some limitations. First of all, the data and the PROMs were collected retrospectively. Secondly, there is not an accurate standardization of the two groups based on age criteria and anaesthesiologic risks. Another limitation is that the postoperative rehabilitation program developed for each patient’s cohort was not considered. Radiographic evaluation during follow-up would have been helpful, but this study was based on clinical evaluations and patient-reported outcome measures. Finally, our results are based on a relatively small cohort of patients, so the reproducibility and the reliability of the procedures should be validated involving a larger study population.

## 5. Conclusions

The results of a comparison between the one-stage procedure and the two-stage procedure of total bilateral knee replacement did not show any clear differences in functionality. Both groups had an improvement in their symptoms and quality of life, with a good degree of satisfaction with daily activities and without the perception that they had undergone surgery.

With the same functional results in both groups, it was therefore useful to shift the comparison on the complications of the two procedures, evaluating both our case studies and the literature.

The one-stage approach in the treatment of bilateral knee arthritis proves to be a valid option that provides patients with a treatment with less subjective stress, thereby reducing healthcare costs. The lack of difference in the occurrence of complications between the two groups supports the idea that with the right indications, the simultaneous procedure can be considered safe. The indication cannot overlook a careful evaluation of the patient’s overall health status, as well as of the surgeon’s confidence in the technique.

## Figures and Tables

**Figure 1 healthcare-12-01902-f001:**
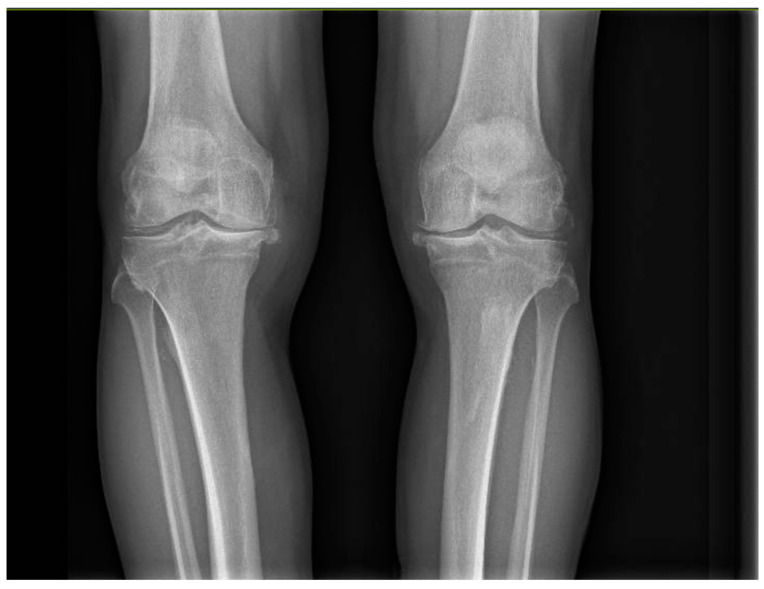
Severe bilateral osteoarthritis of the knee.

**Figure 2 healthcare-12-01902-f002:**
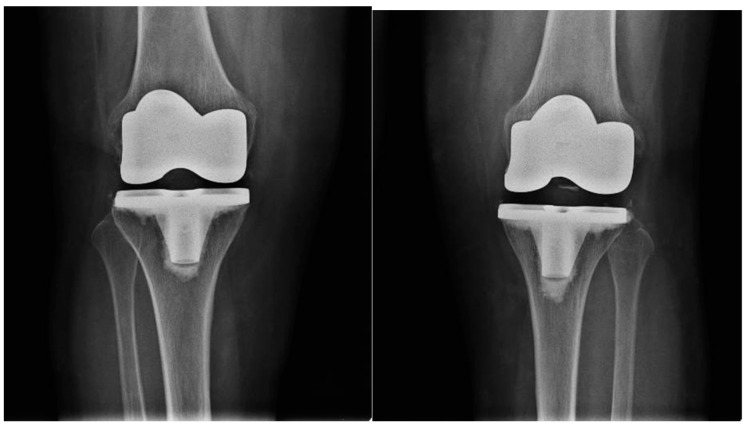
Postoperative radiography of bilateral total knee arthroplasty.

**Figure 3 healthcare-12-01902-f003:**
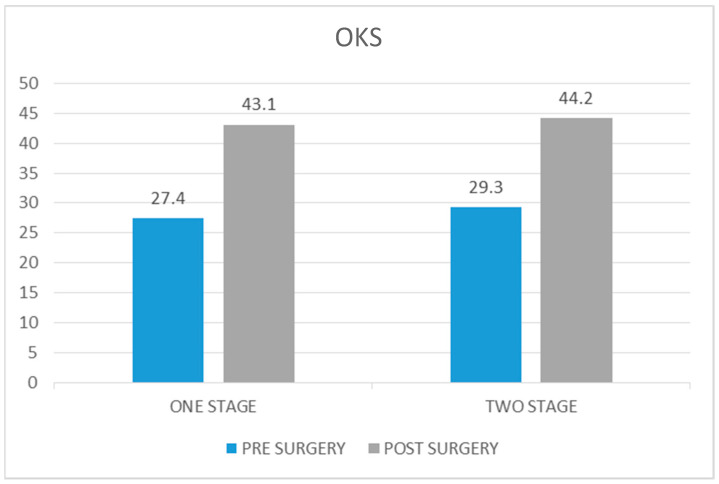
Comparison of Oxford Knee Score pre and post surgery in one-stage procedure vs. two-stage procedure.

**Figure 4 healthcare-12-01902-f004:**
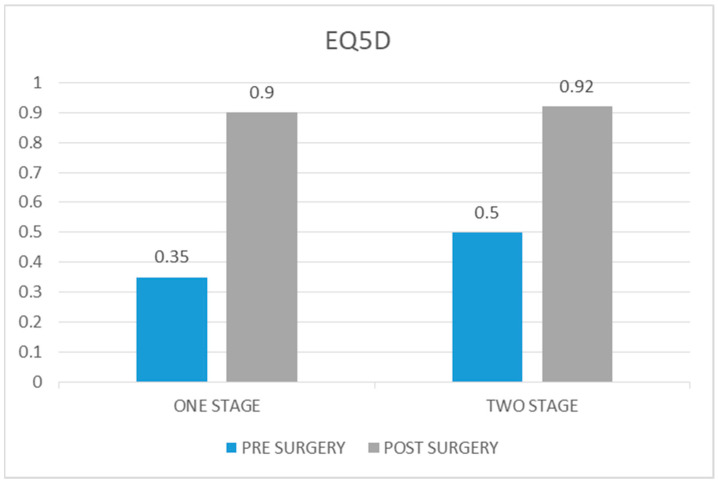
Comparison of EQ5D scores pre and post surgery in one-stage procedure vs. two-stage procedure.

**Figure 5 healthcare-12-01902-f005:**
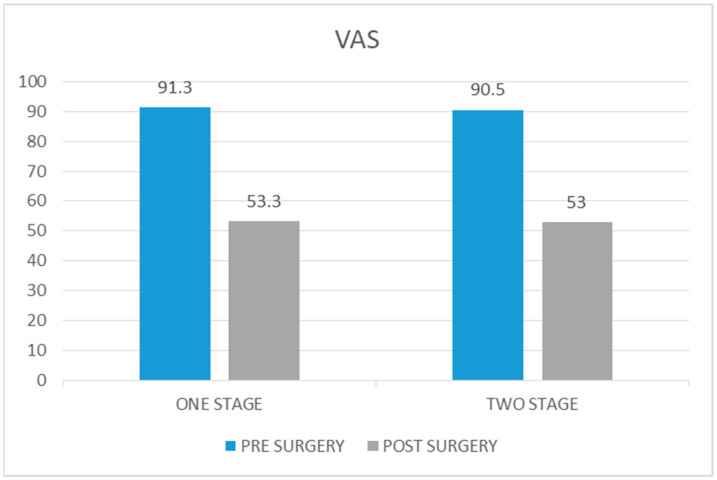
Comparison of Visual Analogue Scale scores pre and post surgery in one-stage procedure vs. two-stage procedure.

**Figure 6 healthcare-12-01902-f006:**
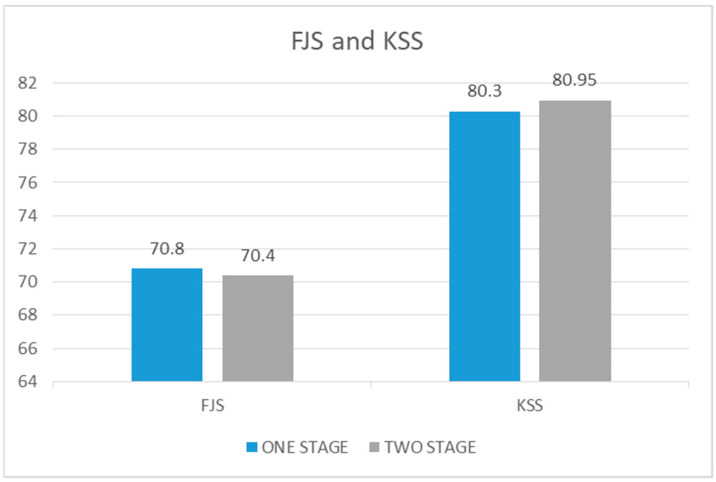
Comparison of FJS and KSS scores pre and post surgery in one-stage procedure vs. two-stage procedure.

**Table 1 healthcare-12-01902-t001:** Demographic characteristics of the population.

Groups	Case Group One-Stage TKA	Control Group Two-Stage TKA	
No. of patients	43	66	
Age	70.2 [42–85]	64.2 [48–75]	0.001219 *p* value < 0.05
Gender	14 M (33%) 29 F (67%)	30 M (45%) 36 F (55%)	0.2201 *p* value > 0.05
Follow-up [months]	59.2 [12–94]	52.8 [12–78]	0.002138 *p* value < 0.05

**Table 2 healthcare-12-01902-t002:** Comparison of the number of patients affected by comorbidities in the one-stage procedure vs. the two-stage procedure.

Comorbidities	Case Group One-Stage TKA	Control Group Two-Stage TKA
Diabetes mellitus	6	8
Hypertension	27	27
Coronary artery disease	7	7
Anxious–depressive syndrome	1	5
Liver disease	1	3
Nephropathy	0	3

## Data Availability

Data are contained within the article.

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
