# Peer review of "Bilateral Total Knee Arthroplasty (TKA) in a One-Stage Procedure Versus Two-Stage Procedure: A Retrospective Study"

_healthcare, 2024, doi:10.3390/healthcare12181902_

Round 1
Reviewer 1 Report
Comments and Suggestions for Authors
Thank you for submitting the article, the idea is up ti date and always interesting. I will suggest just a minor change.
1. Materials and methods section - I would advise you to state which type of implant you used in one and the other group, whether PS or CR implants were used, whether the implants differed by manufacturer, etc.
2. lines 135 to 227 - If you describe in detail the results of one group, and then the control group, and then all the mentioned results are accompanied by tables, I advise you to avoid a detailed description in the text, since all the above is clearly visible in the tables. Please be less detailed in the description of the tables.
3. line 246 - I will suggest to replace the term bag with the term dose, it is more medical language term
Author Response
Thank you for reviewing our article.
- We added the type of prosthesis used.
- We preferred to keep the description of the tables in the text, as it seemed clearer.
- We have replaced the term as indicated.
Reviewer 2 Report
Comments and Suggestions for Authors
1. Expand EQ5D and VAS in the manuscript.
2. Information about OKS, EQ5D, VAS, FJS, and KSS should be included in the manuscript to understand the significance of these measures.
3. The details of any agreements signed with the patients as part of the study is not submitted.
4. On page No.2, line no.73, Figure 1 is missing in the manuscript.
5. On page No.3, lines no.107 Figure 2 have D and S. Why there are two images D and S, and what it denotes should be presented.
6. On pages 6,7,8 and 9, Graphs 1,2,3, and 4 respectively represent dots as commas in case and control group scores. (Ex. In Graph 1 27.4 is given as 27,4. This creates confusion in interpreting the data)
7. Oxford Knee Score (OKS) is represented as OHS on page no.5 and Graph 1.
Comments on the Quality of English LanguageCan be improved
Author Response
Thank you for reviewing our article.
- We have expanded and enhanced the EQ5D and VAS, explaining what they are and how they were added.
- We have expanded and enhanced the PROMs, explaining what they are and how they were added.
- We have corrected the caption of the figure, being able to refer to the radiographic image as Figure 1 also in the text.
- We have eliminated D and S, relative to the laterality in Italian. The radiography shows a bilateral gonarthrosis.
- We are sorry, but we have not been able to correct this error on the decimals for a problem of the system used to make the graphs.
- We corrected it.
- We tried to improved our English
Reviewer 3 Report
Comments and Suggestions for Authors
Dear authors,
Thank you for submitting this interesting paper.
Overall, the research design and outcomes do not match well with the research purpose presented in the introduction.
Furthermore, even taking into accounts of the retrospective study and other limitations, the study results are not sufficiently supported to draw any conclusions
In case you maintain the results of this study, reorganize the research purpose in the introduction and if you keep the research purpose, additional detailed explanations are added or modified in method & material part and result part

Comments on the Quality of English LanguageThere is nothing special to point out about English.
However, some technical terms and literary expressions need to be slightly revised.
If English is not your native language, please attach an English proofreading document by an expert.
Author Response
We thank you for the kindness of your review.
1. Introduction
We tried to change the aim of our study, trying to focus on functional results and collecting studies on possible perioperative complications.
2. Materials and Methods
- We have expanded and enhanced the PROMs, explaining what they are and how they were added.
- No further complications could be investigated due to lack of data collected during patient hospitalization.
- We have added the average surgical time for each procedure. The tourniquette has never been used, as explained in the section on surgical techniques.
3. Results
- We added p-value and static differences between the two groups where possible.
- We have assessed differences by age and gender due to lack of data collected previously.
- We have changed the exposure for control groups and case groups.
- We preferred to keep both the graphical description and textual description of the topics in the tables for clarity.
- Table 2 presented the comorbidities of selected patients.
3.1 Result analisys
- We wrote pvalue in italics.
- We didn’t understand the comment.
- We preferred to keep the description in the section on analysis of results.
- We have preferred to show the differences, even if they are not statistically significant.
- We preferred to keep this exposition because it seemed clear.
- We have assessed differences by age and gender due to lack of data collected previously.
4.Discussion
We have changed the form of the sentence to be as clear as possible.
Reviewer 4 Report
Comments and Suggestions for Authors
The manuscript provides the results of a retrospective study on bilateral TKA in one or two stage procedure. The authors included in the study 43 patients that underwent one-stage bilateral TKA, and 66 patients received two-stage TKA
Strengths:
- Relevance: The topic is highly relevant to the field, addressing a common clinical challenge.
- Comprehensive Literature Search: The authors conducted also a thorough literature search for the discusssection of the mannuscript, including a wide range of studies.
- Structured Analysis: The review is well-structured, with a clear methodology section detailing the search strategy, inclusion and exclusion criteria, and methods for data extraction and synthesis.
Areas for Improvement:
- Language and Clarity: While the manuscript is generally well-written, there are occasional grammatical errors and awkward phrasings.
Conclusion: The manuscript provides a valuable contribution to the literature in the field. With revisions to improve language, the review will be a useful resource for clinicians and researchers in the field. There is a need for large RCTs.
Author Response
Thank you for your kind review. We are pleased that you think our article can contribute to the topic. We tried to improve the exposure in English.
Round 2
Reviewer 3 Report
Comments and Suggestions for Authors
Only part of previous comments are reflected. Conclusions based on the description of the content are not yet clear.
Please see attached comments.

Comments on the Quality of English LanguageThere is nothing special to mention, but if it is not your native language, please submit a certificate of English proofreading.
Author Response
We thank you for your diligence and accuracy in your revisions. We have tried to comply with the corrections as much as possible.
